# Zero-shot CT Field-of-view Completion with Unconditional Generative Diffusion Prior

**Kaiwen Xu**[1]                                                    KAIWEN.XU@VANDERBILT.EDU

**Aravind R. Krishnan**[1]                              ARAVIND.R.KRISHNAN@VANDERBILT.EDU

**Thomas Z. Li**[1]                                          THOMAS.Z.LI@VANDERBILT.EDU

**Yuankai Huo**[1]                                          YUANKAI.HUO@VANDERBILT.EDU

**Kim L. Sandler**[2]                                              KIM.SANDLER@VUMC.ORG

**Fabien Maldonado**[2]                                 FABIEN.MALDONADO@VUMC.ORG

**Bennett A. Landman**[1,2]                          BENNETT.LANDMAN@VANDERBILT.EDU

[1] *Vanderbilt University, 2301 Vanderbilt Place, Nashville, 37235, United States*

[2] *Vanderbilt University Medical Center, 1211 Medical Center Drive, Nashville, 37232, United States*

**Editors:** Under Review for MIDL 2023

## Abstract

Anatomically consistent field-of-view (FOV) completion to recover truncated body sections has important applications in quantitative analyses of computed tomography (CT) with limited FOV. Existing solution based on conditional generative models relies on the fidelity of synthetic truncation patterns at training phase, which poses limitations for the generalizability of the method to potential unknown types of truncation. In this study, we evaluate a zero-shot method based on a pretrained unconditional generative diffusion prior, where truncation pattern with arbitrary forms can be specified at inference phase. In evaluation on simulated chest CT slices with synthetic FOV truncation, the method is capable of recovering anatomically consistent body sections and subcutaneous adipose tissue measurement error caused by FOV truncation. However, the correction accuracy is inferior to the conditionally trained counterpart.

**Keywords:** Field-of-view extension, Denoising diffusion implicit model, Zero-shot learning, Computed tomography

## 1. Introduction

Quantitative analysis of medical images is less effective when body sections of interest are partially truncated by limited imaging field-of-view (FOV). This is especially an issue for opportunistic assessment of body compositions using routine chest computed tomography (CT) (Troschel et al., 2020; Xu et al., 2021; Luo et al., 2021; Xu et al., 2022a). A previous study achieved anatomically consistent FOV extension of chest CT by training a generative model conditioned on synthetic truncation patterns (Xu et al., 2022b). However, the effectiveness of this approach heavily relies on the fidelity of simulated truncation patterns, which makes it difficult to generalize to applications with truncation patterns that are not considered in the simulation. Recent studies have demonstrated the possibility for zero-shot sampling of semantic plausible images conditioned on partially corrupted image data using an unconditionally trained generative diffusion prior (Lugmayr et al., 2022; Fei et al., 2023). The conditioning information is only needed at the inference, making it extremely flexible for applications when corruption patterns are difficult to predict.

In this pilot study, we developed RePaint-DDIM, a variant of the RePaint framework proposed by (Lugmayr et al., 2022) modified for the reverse sampling scheme of denoising diffusion implicit models (DDIM) (Song et al., 2021) and evaluated the method for the task of FOV completion of routine lung screening low-dose CT.

## 2. Method

A generative diffusion model is trained to reverse a forward Markovian diffusion process that progressively turns an image into noise. Once developed, the model is capable of sampling a realistic image from random noise following a recurrent procedure. Compared to the original sampling scheme of denoising diffusion probabilistic model (DDPM) (Ho et al., 2020), the DDIM reverse sampling scheme turns the generative process into a deterministic process and reduces the required sampling steps with a shortened sampling trajectory (Song et al., 2021).

To condition the reverse sampling process on known image regions, RePaint (Lugmayr et al., 2022) alters the process by iteratively replacing known regions of the intermediate reverse sampled image with forward sampled image at each step of the DDPM denoising process. To address the disharmony of the generated parts of the image, an additional resampling process is introduced by iterating forward diffusion, known region replacing, and denoising between adjacent sampling steps. In this study, we modified the RePaint

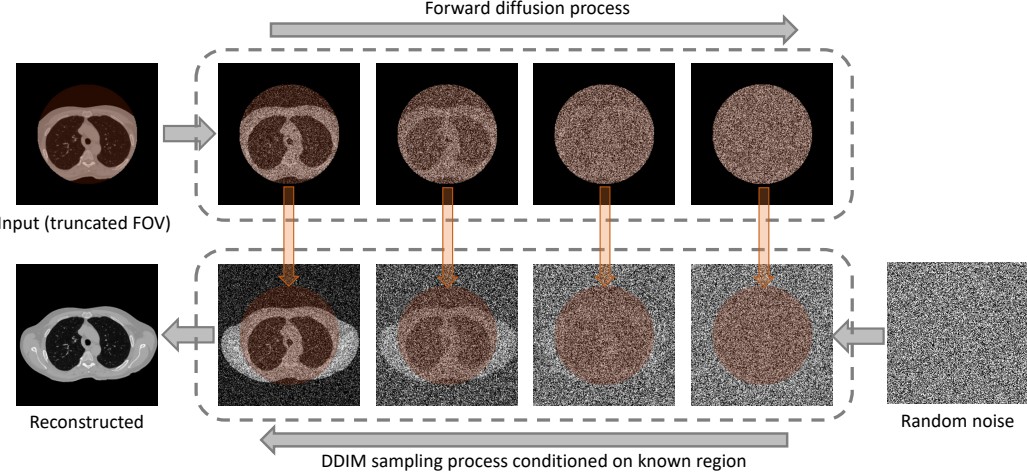

Figure 1: Overview of the mechanism for conditional field-of-view completion based-on unconditionally pre-trained generative diffusion prior.

algorithm for the DDIM sampling process to take advantage of the shortened inference time. In the resampling steps, instead of iterating between adjacent denoising steps, the modified algorithm iterates between the predicted fully denoised image ($x_0$) and intermediate sampled noisy image ($x_t$) in each DDIM sampling step. We called this modified version as RePaint-DDIM. An overview of the sampling workflow is demonstrated in Figure 1. Detailed steps are provided in Algorithm 1.

---

**Algorithm 1:** RePaint-DDIM sampling algorithm for CT field-of-view completion.

---

**Input:** $\tilde{x}_0$, CT slice with FOV truncation; $m$, FOV region mask; $\alpha_0 : \alpha_T \in (0, 1]$, diffusion
schedule; $\epsilon_\theta$, pretrained denoising model.
**Output:** $x_0$, CT slice with completed FOV
$x_T \leftarrow \mathcal{N}(0, I)$;
**for** $t \leftarrow T$ **to** 1 **do**
  **for** $u \leftarrow 1$ **to** $U$ **do**
    $\hat{x}_0 \leftarrow \frac{1}{\sqrt{\alpha_t}} \left( x_t - \sqrt{1 - \alpha_t} \epsilon_\theta(x_t; t) \right)$;
    $\hat{x}_0 \leftarrow m \odot \tilde{x}_0 + (1 - m) \odot \hat{x}_0$;
    **if** $u < U$ **then**
      $\epsilon \leftarrow \mathcal{N}(0, I)$;
      $x_t \leftarrow \sqrt{\alpha_t} \hat{x}_0 + \sqrt{1 - \alpha_t} \epsilon$;
    **end**
  **end**
  $x_{t-1} \leftarrow \sqrt{\alpha_{t-1}} \hat{x}_0 + \sqrt{1 - \alpha_{t-1}} \epsilon_\theta(x_t; t)$;
**end**

---

## 3. Experiment and Discussion

We pretrained an unconditional DDPM using 71,319 lung cancer screening low-dose CT slices with complete body in FOV. Details of the collection of this dataset were provided in (Xu et al., 2022b). Slices were resized to $256 \times 256$ and clipped to HU range $[-1000, 600]$. The model was trained with diffusion steps $T = 1000$, linear beta scheduler, and a batch size of 24. The model was trained for 30,000 iterations. At inference, we use 50 denoising steps and 20 resampling steps for each denoising step.

We evaluated the RePaint-DDIM on 2,657 simulated FOV truncation slices generated from 145 withheld slices with complete body in FOV. The anatomical consistency of the synthetic body sections was quantitatively evaluated by the agreement of subcutaneous adipose tissue (SAT) measured on reconstructed slices with the same measurement on untruncated version following the method used in (Xu et al., 2022b). We compared the RePaint-DDIM with a conditionally trained model as developed in (Xu et al., 2022b) (termed S-EFOV). The results are provided in Figure 2. The method is capable of restoring anatomical consistent body sections in the truncated region and correct the measurement error of SAT. However, the correction accuracy is inferior to the conditionally trained counterpart.

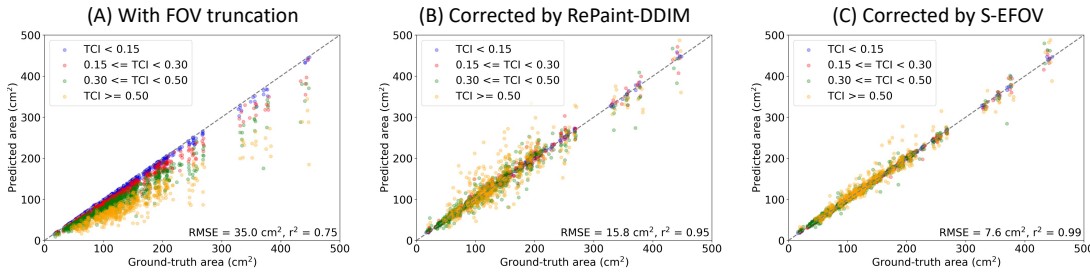

Figure 2: Evaluation of the anatomical consistency of the field-of-view (FOV) completion results. Tissue truncation index (TCI) reflects the severity of synthetic FOV truncation.

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
