# OpenReview forum: "Zero-shot CT Field-of-view Completion with Unconditional Generative Diffusion Prior"
_MIDL.io/2023/Short_Paper_Track — MIDL 2023 Short paper track Poster_

### Official Review · Reviewer_FN3u · 2023-04-12
**Interesting approach, promising preliminary results**

**Rating:** 8
**Confidence:** 4

**Review:**

The authors present the preliminary results they obtained with a new approach they proposed to complete the FOV of routine lung screening low-dose CT. The approach is a diffusion model variant whose training is not conditioned on the final task.
- \+ : The approach is interesting and seems innovative.
- \+ : The validation, which relies on a large number of samples and two evaluation criteria, appears solid.
- \+ : The results seem promising.
- \- : For now the approach performs less well than a conditionally trained model.

---

### Official Review · Reviewer_rXDP · 2023-04-25
**Anatomically consistent field-of-view (FOV) completion in CT using RePaint**

**Rating:** 6
**Confidence:** 4

**Review:**

This short paper uses two methods from literature to solve the problem of anatomically consistent field-of-view (FOV) completion to recover truncated body sections in CT images with limited FOV in order to be able to perform  subcutaneous adipose tissue measurement. The first method the authors use is the RePaint framework (Lugmayr et al., 2022), based on a pretrained unconditional generative  denoising diffusion probability model (DDPM), which can generate images to fill truncated parts of the images with arbitrary truncation masks at inference phase and the second is diffusion implicit models (DDIM) (Song et al., 2021) which shortens inference time of DDPMs. The results show that the proposed method’s accuracy was inferior to a conditionally trained method. However, the proposed method has the advantage of recovering arbitrarily truncated CTs. It seems more future work is needed to improve the performance of the proposed method